# Trends in Participation, Sex Differences and Age of Peak Performance in Time-Limited Ultramarathon Events: A Secular Analysis

**DOI:** 10.3390/medicina58030366

**Published:** 2022-03-01

**Authors:** Mabliny Thuany, Thayse Natacha Gomes, Elias Villiger, Katja Weiss, Volker Scheer, Pantelis T. Nikolaidis, Beat Knechtle

**Affiliations:** 1Centre of Research, Education, Innovation and Intervention in Sport (CIFI2D), Faculty of Sports, University of Porto, 4200-450 Porto, Portugal; mablinysantos@gmail.com; 2Department of Physical Education, Federal University of Sergipe (UFS), Sao Cristovao 49100-000, Brazil; thayse_natacha@hotmail.com; 3Klinik für Allgemeine Innere Medizin, Kantonsspital St. Gallen, 9000 St. Gallen, Switzerland; evilliger@gmail.com; 4Medbase St. Gallen Am Vadianplatz, 9001 St. Gallen, Switzerland; katja@weiss.co.com; 5Ultra Sports Science Foundation, 109 Boulevard de l’Europe, 69310 Pierre-Benite, France; volkerscheer@yahoo.com; 6School of Health and Caring Sciences, University of West Attica, 12243 Athens, Greece; pademil@hotmail.com; 7Institute of Primary Care, University of Zurich, 8091 Zurich, Switzerland

**Keywords:** ultra-marathon, age of peak, performance, endurance

## Abstract

*Background and Objectives*: Increases in the number of participants in time-limited ultra-marathons have been reported. However, no information is available regarding the trends in participation, performance and age in 12 h and 24 h time-limited events. The aim of the study was to describe the trends in runners’ participation, performance and age in 12 h and 24 h ultra-marathons for both sexes and to identify the age of peak performance, taking into account the ranking position and age categories. *Materials and Methods*: The sample comprised 210,455 runners in time-limited ultra-marathons (female 12 h = 23,706; female 24 h = 28,585; male 12 h = 61,594; male 24 h = 96,570) competing between 1876 and 2020 and aged 18 to 86 years. The age of peak performance was tested according to their ranking position (first–third; fourth–tenth and >tenth position) and taking into account their running speed in different age categories (<30 years; 31–40 years; 41–50 years; 51–60 years; >60 years), using the Kruskal–Wallis test, followed by the Bonferroni adjustment. *Results*: An increase in the number of participants and a decrease in running speed were observed across the years. For both events, the sex differences in performance decreased over time. The sex differences showed that male runners performed better than female runners, but the lowest differences in recent years were observed in the 24 h ultra-marathons. A positive trend in age across the years was found with an increase in mean age (“before 1989” = 40.33 ± 10.07 years; “1990–1999” = 44.16 ± 10.37 years; “2000–2009” = 45.99 ± 10.33 years; “2010–2020” = 45.62 ± 10.80 years). Male runners in 24 h races were the oldest (46.13 ± 10.83 years), while female runners in 12 h races were the youngest (43.46 ± 10.16 years). Athletes ranked first–third position were the youngest (female 12 h = 41.19 ± 8.87 years; female 24 h = 42.19 ± 8.50 years; male 12 h = 42.03 ± 9.40 years; male 24 h = 43.55 ± 9.03 years). When age categories were considered, the best performance was found for athletes aged between 41 and 50 years (female 12 h 6.48 ± 1.74 km/h; female 24 h 5.64 ± 1.68 km/h; male 12 h 7.19 ± 1.90 km/h; male 24 h 6.03 ± 1.78 km/h). *Conclusion*: A positive trend in participation in 12 h and 24 h ultra-marathons was shown across the years; however, athletes were becoming slower and older. The fastest athletes were the youngest ones, but when age intervals were considered, the age of peak performance was between 41 and 50 years.

## 1. Introduction

Ultra-marathon races are running events where distances are longer than the traditional marathons and can be distance-(50 km, 100 km, 50 miles and 100 miles ultra-marathons) or time-limited (e.g., 6 h, 12 h, 24 h, 48 h) [1]. Ultra-marathons date back to 1861, when the American Edward Payson Weston walked 713 km from Boston to Washington, DC, USA. Over the next 14 years, a competition to find the world’s best pedestrian was observed between the United States and England [2]. However, only in 1988, the International Association of Ultrarunners was officially recognized (www.worldathletics.org/disciplines/ultra-running/ultra-running; accessed on 6 January 2022).

Previous studies have shown that the number of participants in these events has increased worldwide in recent years [3]. Data covering ~85% of the ultra-marathons held worldwide during 1996–2018 showed an increase of 1676% in runner participation [4]. A recent study investigated 369,969 men and 69,668 women competing between 1960 and 2019 in 100 km ultra-marathon races, showing a positive trend in athlete participation during the studied time [5]. Similar results were found in 6 h [6], 161 km and 100-mile events [5,7,8].

This increase is explained, partially, by the increased numbers of participation of young [9,10], female [7] and elderly runners [5]. However, the increase in the number of ultra-marathon participants is not followed by performance improvement, that is, the athletes have become slower over the years [5]. For example, data covering 5,010,730 results from 15,451 ultra-marathon running events showed that runners were 15% slower compared to those from 1996 [4]. Previous studies indicated that morphological characteristics, such as body fat and body mass index [11], pacing strategy [12], training experience [13] and age [14] were associated with ultra-marathon performance, especially in distance-based events.

Considering the role of biological characteristics in ultra-marathon performance [11], the age of peak performance was previously investigated. In general, the age interval was found to be between 30 and 50 years in the “Swiss Alpine Marathon”, 50 km, and 161 km [14,15,16]. However, different results were found when considering time-limited events (i.e., 12 h and 24 h), where the age of the best ranked athletes was between 38 and 45 years [17]. This is an invaluable information for both coaches and athletes to prepare training for ultra-endurance events. This information can also be relevant for older athletes, due to the decreases in VO_2max_, lactate threshold velocity, blood volume and muscle mass [18].

Few studies were conducted aiming to understand the trends in performance and age in time-limited ultra-marathons. Most of them investigated the pacing strategy, taking into account the predictors of the performance and sex differences, with limited temporal gap [17,19,20]. Since knowing the athlete’s profile and temporal trends could guide the long-term training, the purposes of this study were (i) to describe trends in runners’ participation, age and performance in 12 h and 24 h ultra-marathon races for both sexes; and (ii) to identify the age of peak performance, taking into account the ranking position and age categories. Based on previous research, we hypothesized that there is (i) an increase in participants across the years and a decrease in performance; and (ii) the age of peak performance would be between 40 and 50 years.

## 2. Materials and Methods

### 2.1. Ethical Approval

The institutional review board of St Gallen, Switzerland, approved this study (EKSG 1 June 2010). Since the study involved the analysis of publicly available data, the requirement for informed consent was waived.

### 2.2. Design and Sample

This is an exploratory study, using information obtained from the event’s official webpages. Data were collected by one of the authors (E.V.) from the website of “Deutsche Ultramarathon-Vereinigung” (https://www.d-u-v.org/; accessed on 6 January 2022). All information derives from the official available results for female and male participants in 12 h and 24 h ultra-marathon races (https://statistik.d-u-v.org/geteventlist.php; accessed on 6 January 2022) (female 12 h = 23,706; female 24 h = 28,585; male 12 h = 61,594; male 24 h = 965,570), between 1876 and 2020 for male, and between 1971 and 2020 for female runners. In these races, the participants have to run as many kilometers as possible for 12 h or 24 h, depending upon the race they start in.

The available information included the year of the event, the athletes’ name, date of birth, sex, ranking, average running speed, completed distance and country of residence. The athletes’ age was computed, taking into account the year of birth and the year of the competition. Age range was 18–80 years, for both sexes, in 24 h, and 12 h for female athletes. For 12 h male athletes, the age range was 18–86 years. For the present study, athletes were analyzed regarding age categories (<30 years of age; 31–40 years of age; 41–50 years of age; 51–60 years of age; >60 years of age) and ranking position (first–third; fourth–tenth; >tenth). Athletes with incomplete information and aged below 18 years were excluded.

### 2.3. Statistical Analysis

We computed descriptive statistics, including percentages, means and standard deviations. Data normality was tested using the Kolmogorov–Smirnov test by sex and event. To identify ultra-marathoners’ differences according to age, running speed and achieved distance, a multivariate analysis of variance was conducted, and Pillai’s trace values were considered, given that variance and covariance homogeneity were not observed. Eta squared (*n*^2^) was used as a measurement for the effect size. To analyze trends in performance and age, all the athletes were considered by sex, and they were split based on the events’ year into four groups (”<1900”; “1901–1970”; ”1971–1989”; “1990–1999”; “2000–2009”; “2010–2020”), when information was available.

Sex differences in running speed (km/h)—which was considered as a performance indicator—were calculated, and results were presented in delta, with a positive value indicating male higher performance. Age of peak performance was verified through two approaches: (1) considering the athletes’ ages, according to their ranking position (first–third; fourth–tenth and >tenth position); (2) taking into account their running speed (km/h) into the different age categories (<30 years; 31–40 years; 41–50 years; 51–60 years; >60 years). For both approaches, the Kruskal–Wallis test was used to estimate significant differences, followed by the Mann–Whitney test (with *p*-adjustment for the number of comparisons), to identity where the differences were observed. Statistical analysis was performed in the software SPSS 26.0, considering *p* < 0.05.

## 3. Results

The sample comprised 210,455 runners, who completed a 12 h and 24 h ultra-marathon from 1876 to 2020 (female 12 h = 23,706; female 24 h = 28,585; male 12 h = 61,594; male 24 h = 96,570).

### 3.1. Participation Trend in Ultra-Marathons

Figure 1 presents the trend in athletes’ participation in 12 h and 24 h ultra-marathons for both sexes. In general, for both sexes and events, an increase in the number of participants over the years was observed, with a greater increase during the last decade, especially for male runners. However, in 2020, a decrease in the number of runners was observed due to the COVID-19 pandemic, which led to the reduction of sports events all over the world.

### 3.2. Trends in Performance and Sex Differences

Trends in performance and sex differences in performance are presented below (Figure 2 and Figure 3). For both sexes and events, a decrease in running speed was observed across the years. Considering the first and the last year of participation, the highest difference in running speed was found for male athletes in the 12 h events (−4.65 km/h), while the lowest difference was observed for male athletes in the 24 h events (−1.37 km/h). Sex differences in performance are presented in Figure 3. For both events, sex differences in performance decreased over time. In general, male runners performed better than female runners, but the lowest differences in recent years were observed for 24 h ultra-marathons, where female runners performed better than their male peers in three years (1986, 2003 and 2006). On the other hand, during the first years, the highest sex differences in performance were also observed among athletes in the 24 h ultra-marathons, favoring male runners.

### 3.3. Trends in Age over the Years

Changes in age across the years are presented in Figure 4 for both sexes. When considering the total sample, regardless of sex and/or time-limited ultra-marathons, a positive trend in age across the years was found with an increase and stabilization in mean age (“<1989” = 40.33 ± 10.07 years; “1990–1999” = 44.16 ± 10.37 years; ”2000–2009” = 45.99 ± 10.33 years; “2010–2020” = 45.62 ± 10.80 years).

The highest mean age was observed for male runners in 24 h ultra-marathons (46.13 ± 10.83 years), while the lowest value was found for female runners in 12 h ultra-marathons (43.46 ± 10.16 years). Except for female runners in 12 h ultra-marathons, the mean age for all the groups was higher than 40 years after the 1980s (Figure 4).

Results for multivariate analysis are presented in Table 1. Significant inter-group differences were verified for all variables ((Pillai’s trace = 0.940); F _(8840)_ = 32,006.115, *p* < 0.001; *n*^2^ = 0.313). Effect size indicated that 31% of inter-group differences in the variables were related to the sex of the participants and event. For both events, the highest mean age, as well as the best mean performance and the greatest mean distance, were observed for male runners.

### 3.4. Age of Peak Performance by Ranking Position

The results for the age of peak performance are presented in Figure 5, for both sex and event. Statistically significant differences were found for both sexes and events when comparing ranking positions (female 12 h − H_(2)_ = 164.76; *p* < 0.001; female 24 h − H_(2)_ = 200.57; *p* < 0.001; male 12 h − H_(2)_ = 894.51; *p* < 0.001; male 24 h − H_(2)_ = 760.88; *p* < 0.001), where those classified between the first and third positions were younger (female 12 h = 41.19 ± 8.87 years; female 24 h = 42.19 ± 8.50 years; male 12 h = 42.03 ± 9.40 years; male 24 h = 43.55 ± 9.03 years) than their peers who ranked between the fourth and tenth (female 12 h = 42.52 ± 9.73 years; female 24 h = 43.73 ± 9.11 years; male 12 h = 44.77 ± 10.64 years; male 24 h = 45.45 ± 10.17 years), and >tenth positions (female 12 h = 43.97 ± 10.36 years; female 24 h = 45.26 ± 10.33 years; male 12 h = 46.07 ± 11.12 years; male 24 h = 46.67 ± 11.16 years).

### 3.5. Ultra-Marathoners’ Performance by Age Categories

Descriptive information, regarding age categories, indicated the highest frequency of athletes aged 41–50 years (36.9%), followed by those aged 31–40 years (24.9%), 51–60 years (21.6%), >60 years (8.6%), and ≤30 years (8.0%). Significant differences for age categories in running speed were observed between almost all groups, for both sexes and events (Figure 6) (female 12 h − H_(4)_ = 471.28; *p* < 0.001; male 12 h − H_(4)_ = 1407.59; *p* < 0.001; female 24 h − H_(4)_ = 420.28, *p* < 0.001; male 24 h − H_(4)_ = 1643.42; *p* < 0.001), except between male runners aged “≤30 years” and “51–60 years” in the 12 h ultra-marathons. On average, the highest running speed was found for athletes aged 41–50 years (female 12 h 6.48 ± 1.74 km/h; female 24 h 5.64 ± 1.68 km/h; male 12 h 7.19 ± 1.90 km/h; male 24 h 6.03 ± 1.78 km/h).

## 4. Discussion

The purposes of this study were (i) to describe the trends in runner participation for both sexes, performance and age in 12 h and 24 h ultra-marathon races and (ii) to identify the age of peak performance, while taking into account the ranking position and age categories. The main findings were: (i) male athletes competed more often than female athletes in these events; (ii) male runners presented the best performance, but a decrease in the gap between sexes was observed during the last years; (iii) the highest median age was found for male athletes in 24 h ultra-marathons, while female athletes in 12 h ultra-marathons showed the lowest median age; (iv) athletes who ranked between the first and third positions were the youngest ones in both sexes and races (female 12 h = 41.19 ± 8.87 years; female 24 h = 42.19 ± 8.50 years; male 12 h = 42.03 ± 9.40 years; male 24 h = 43.55 ± 9.03 years); and (v) considering the age groups, athletes aged between 41 and 50 years were the fastest ones in both sexes.

### 4.1. More Athletes, but Slower

The first important finding was a positive trend in participation for both sexes and events, across the years, which confirms the study hypothesis. Participation has been increasing since 2010, with a decrease in 2020 due to the pandemic situation caused by SARS-CoV-2 [21]. Similar results were previously shown [5,13]. Increases in ultra-marathon participation are associated with a plethora of factors, such as increases in the number of ultra-marathon events; athletes’ “migration” from marathon to ultra-marathon; an increase in participation of female runners, as well as of both younger and older runners [16]. Thus, a report conducted between 2001 and 2018, encompassing 5 km, 10 km, half-marathon, marathon and ultra-marathon events, found that 41.02% of ultra-marathoners participated in more than one race event in 2018, while for 5 km, 10 km, half-marathon and marathon, the values were 33.31%, 16.53%, 22.62% and 17.60%, respectively [4].

The hypothesis related to the decrease in performance was confirmed. The highest decrease was observed among male 12 h runners, while the lowest decrease was found in male 24 h athletes. The decrease in performance across time was previously shown among athletes competing in the “100 km Lauf Biel”, which is the oldest ultra-marathon in the world [22], as well as among time-limited ultra-marathoners (i.e., 6 h, 12 h, 24 h, 48 h, 72 h, 144 h and 240 h) [23].

These decreases in performance trends can be associated with changes in the runners’ profiles. In summary, among non-athletes, running events have become a leisure/social activity, with a reduction of the competitive perspective generally observed in the past [24,25,26]. In addition, the increase in the number of women, as well as younger and older athletes’ participation in recent years, can also be associated with this observed decrease in performance [27,28]. The present results are different from those shown by Teutsch et al. [17], in a study with athletes who completed the 12 h and 24 h races in Basel (Switzerland). The authors reported that between 1988 and 2012, the running performance was stable for both sexes, across the studied years. Similarly, in the “100 km Lauf Biel” (Switzerland), running performance did not significantly change during a 12-year period (1998–2010) [29]. Differences for these results can be related to differences in time interval investigated, as well as the races considered, given that the present study considered different 12 h and 24 h events in the last 140 years.

### 4.2. Getting Older, but the Youngest Were the Fastest

Regarding the age trend, the highest median age values were verified in recent decades (“<1989” = 40.33 ± 10.07 years; “1990–1999” = 44.16 ± 10.37 years; “2000–2009” = 45.99 ± 10.33 years; “2010–2020” = 45.62 ± 10.80 years). So, these two observations (i.e., a decrease in performance and an increase in median age values) may be related. Aging has been inversely associated with running performance due to several factors, such as biological (e.g., a decline in VO_2_max and heart rate frequency) [18] and behavioral changes (e.g., a reduction in exercise training intensity and session duration, changes in nutritional habits and a decrease in training commitment) [18].

Athletes ranked between first and third positions were the youngest ones (female 12 h = 41.19 ± 8.87 years; female 24 h = 42.19 ± 8.50 years; male 12 h = 42.03 ± 9.40 years; male 24 h = 43.55 ± 9.03 years), when compared against those ranked between the fourth and tenth, and >tenth positions. Similarly, the hypothesis was confirmed, given that athletes aged between 41 and 50 years were the fastest ones (female 12 h 6.48 ± 1.74 km/h; female 24 h 5.64 ± 1.68 km/h; male 12 h 7.19 ± 1.90 km/h; male 24 h 6.03 ± 1.78 km/h). These approaches were previously considered to investigate the peak performance [30]. In the present study, they were used due to the fact that one-third of the runners were between 41 and 50 years old, where a simple mean/median age comparison could yield biased results.

Similar results were found by Teutsch et al. [17], where the age of peak performance was achieved between 38 and 45 years. Different from our results, a previous study investigating 35,956 runners who competed between 1998 and 2011 in 100-mile ultra-marathons, found an age of peak performance of 39.2 ± 6.2 years for women and 37.2 ± 6.1 years for men [31]. In addition, in a study aiming to understand the age of peak performance in time-limited ultra-marathons (i.e., 6 h, 12 h, 24 h, 48 h, 72 h, 144 h and 240 h), Knechtle et al. [23] showed that the lowest age of peak performance was found among the 6 h event runners (33.7 years), while the highest age was found among the 48 h event runners (46.8 year). Possible explanations for the differences observed may be related to methodological differences (e.g., temporal interval and sample), as well as athletes’ experience, since the longer the event’s duration, the higher the athletes’ mean/median age [32].

### 4.3. Limitation and Strength of the Study

The first limitation of the present study is the missing data in some years, for both sexes. In addition, it is also relevant to point out the lack of information regarding the environmental characteristics (e.g., wind, altimetry, terrain), that can impair the athletes’ performance across different races [31]. These aspects are important and should be considered in future studies, given their influence on results regarding the trends in athletes’ participation and performance. Secondly, the increase in age across the years can be associated with the aging of the general population [33], especially among those from high-income countries, where most of the running events are performed. Further, increases in the size of worldwide population can be related to increases in the number of runners, and this adjustment was not performed in the present study, since we were not able to find this information alongside each year. Thirdly, the lack of information regarding athletes’ experience (e.g., number of ultra-marathons, training characteristics) highlights that caution must be taken regarding the generalization of the results. We could also not differentiate between road and trail races with elevation changes [1]. On the other hand, to the best of the authors’ knowledge, this is the first study with the purpose to investigate trends in participation, age and performance, as well as peak performance in runners, for both sexes, among competitors who completed 12 h and 24 h time-limited ultra-marathons. Information regarding the peak performance can guide the long-term planning, especially among those athletes who reach peak performance in a marathon and are looking for new challenges in ultra-marathon events.

## 5. Conclusions

In summary, in 12 h and 24 h ultra-marathons held between 1876 and 2020, an increase in athletes’ participation was found for both sexes. Men were faster than women, but decreases in sex differences in performance were shown in recent years. An increase in median age values was observed across the years, with 24 h male runners being the oldest, while the 12 h female runners were the youngest. Considering athletes’ position, the fastest athletes were the youngest ones, but when the age intervals were considered, those aged between 41 and 50 years achieved the best performance.

## Figures and Tables

**Figure 1 medicina-58-00366-f001:**
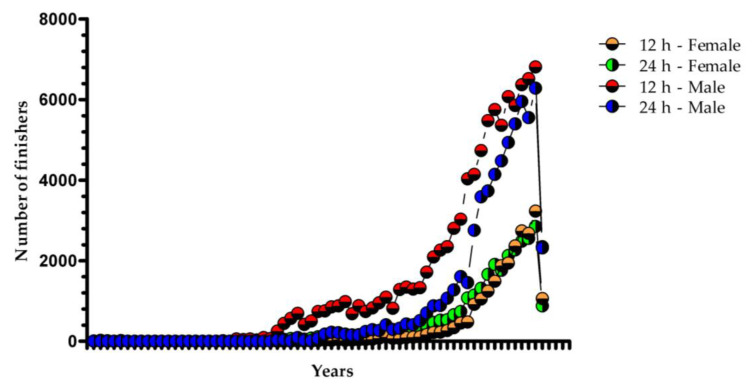
Trends in runners’ participation in 12 h and 24 h ultra-marathons between 1876 and 2020 for both sexes.

**Figure 2 medicina-58-00366-f002:**
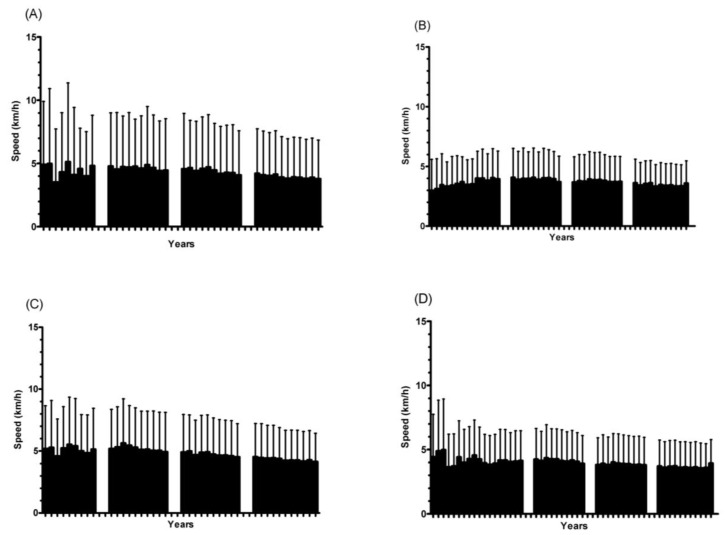
Running speed for ultra-marathoners of both sexes and events ((**A**) female 12 h; (**B**) female 24 h; (**C**) male 12 h; (**D**) male 24 h).

**Figure 3 medicina-58-00366-f003:**
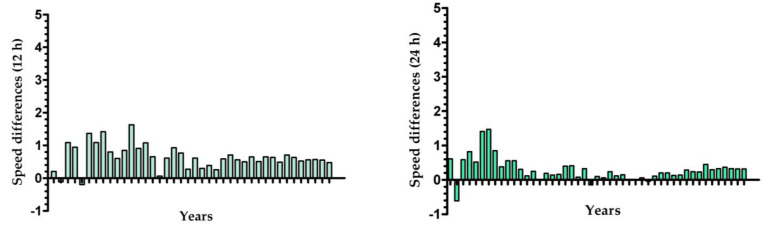
Sex differences in the performance in 12 h and 24 h ultra-marathons.

**Figure 4 medicina-58-00366-f004:**
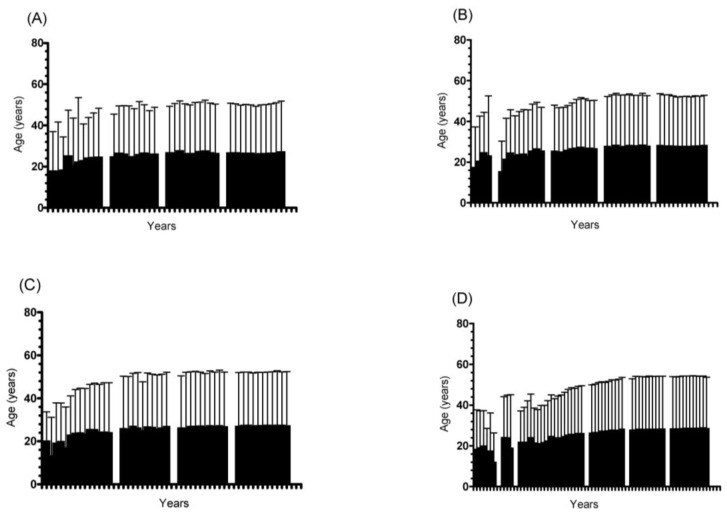
Trends in mean age across the years. ((**A**) 12 h female runners; (**B**) 24 h female runners; (**C**) 12 h male runners; (**D**) 24 h male runners).

**Figure 5 medicina-58-00366-f005:**
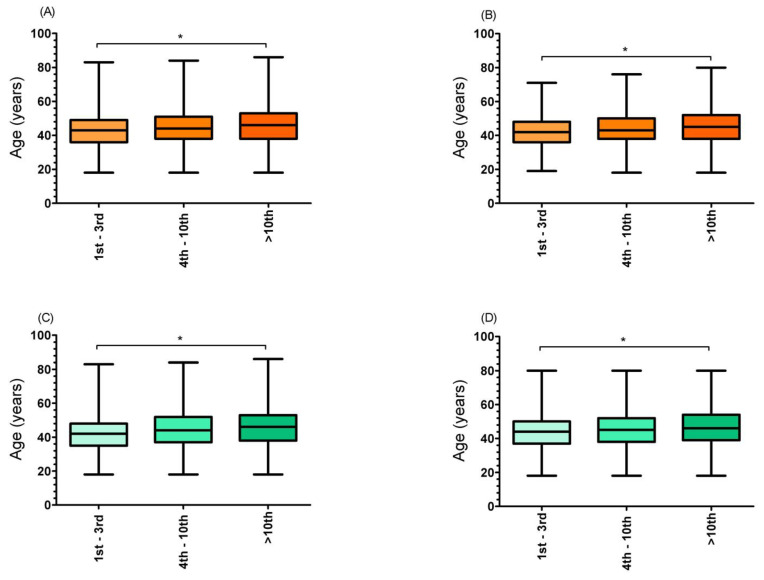
Age of peak performance, by ranking position. ((**A**) 12 h female runners; (**B**) 24 h female runners; (**C**) 12 h male runners; (**D**) 24 h male runners) * Significant differences between all groups.

**Figure 6 medicina-58-00366-f006:**
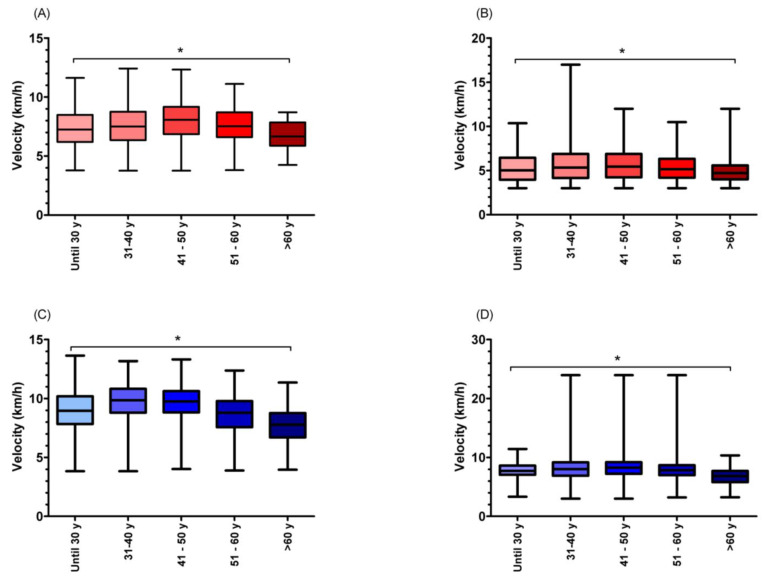
Differences in performance according to age groups. ((**A**) 12 h female runners; (**B**) 24 h female runners; (**C**) 12 h male runners; (**D**) 24 h male runners). * Significant differences between age categories.

**Table 1 medicina-58-00366-t001:** Multivariate analysis results of inter-group differences.

Variables	Female 12 h	Male 12 h	Female 24 h	Male 24 h	*p*-Value	*n* ^2^
Age (years)	43.46 (10.17)	45.29 (10.91)	44.83 (10.07)	46.13 (46.13)	<0.001	0.01
Distance (km)	75.78 (20.36)	83.94 (22.53)	131.70 (39.17)	140.12 (40.98)	<0.001	0.40
Speed (km/h)	6.31 (1.70)	6.99 (1.88)	5.49 (1.72)	5.844 (1.84)	<0.001	0.09

MANOVA Test ((Pillai’s trace = 0.940); F_(8840)_ = 32,006.115, *p* < 0.001; *n*^2^ = 0.313).

## Data Availability

Data are available from the authors upon request.

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
