# Peer review of "Trends in Participation, Sex Differences and Age of Peak Performance in Time-Limited Ultramarathon Events: A Secular Analysis"

_medicina, 2022, doi:10.3390/medicina58030366_

Round 1
Reviewer 1 Report
Overall well done article. Picking 12& 24 races is a novel approach to look at performance characteristics. Comments below
Couple of overall comments:
Is it fair to compare men & women given the date ranges for both groups are so different? For women you are looking at a 49 year span and for men you are looking at a 144 year span. Why not just look at data for both over the past 50 years?
You also may not have data on this but could drop out rate/completion rates affect this data. It could be that while finishing times are getting slower, more people are completing the events.
Abstract: "the sex differences in performance decreased in the last years, when compared to the 32 sex differences in performance during the first years"- could you be more specific with years. Or maybe just say that performance decreased over time?
Male runners in 24-hour races were the oldest 37 (46.13±10.83 years), while female runners in the 12-hour races were the youngest (43.46±10.16 years)- Given that range, I'm assuming this wasn't statistically significant. From a reader perspective it's interesting but essentially they are pretty similar age ranges
Main Paper:
Intro
"This is an invaluable information for both coaches and athletes to prepare training for ultra-endurance events. This information is also crucial for older athletes, due to the decreases in VO2max, lactate threshold velocity, blood volume, and muscle mass". While this is true, I'm not sure how this is "crucial". Most older ultrarunners know they are getting a bit slower
"between 1876 and 95 2020 for male"- just a comment- I've never reviewed/read a paper that had data from 1876. Fantastic!
Results:
Figure 1- Really hard to read given the different dots with same colors. Any reason you can't just do 4 different colored lines?
So the number of people doing these events is increasing. But given you are going all the way back to 1876, the population was much smaller then. In 2021, more people are doing everything. If this were looking at the cost of entering races, you'd have to adjust for inflation. Should you be adjusting for population?
"For both events, sex differences in performance decreased in the last years, when compared to the sex differences in performance during the first years" Same as abstract comment
Figure 2/3- Seems like the X axis should include the years. Also given the men are over a bigger age range, when I look at the graphs, the differences in the x-axis sort of skew directly comparing men's/women's graphs
Discussion:
"In summary, running events have become a leisure/social activity, with a reduction of the competitive perspective generally observed in the past"- I'm not sure about this. I think in the road marathon situation we've seen this with a big split occurring in the early 90's (the Oprah Winfrey effect in the US). Is there really data that over the past 20-30 years, ultrarunners have become more leisure/social activity? I my experience, I just see more runners getting tired of running on the roads competitively and then they head to the trails. I'd also argue that the depth of the field in elite running has gotten much deeper in the past 10-15 years.
I also would consider thinking about being careful with younger/older categorizations in this paper. (For example_ Athletes ranked in 1st–3 rd positions were the youngest ones (female 12-hour = 260 41.19±8.87 years; female 24-hour = 42.19±8.50 years; male 12-hour = 42.03±9.40 years; male 261 24-hour = 43.55±9.03 years)) Really these aren't young runners in the grand scheme of runners.
Regarding course/other factors-we are starting to wonder at some of our races if the fastest times may be a thing of the past due to global warming and warmer conditions.
With courses- were the early races on tracks? Do you have a way to separate out results based on the course. While ultrarunning in general is coming up with more and more difficult races, it seems like most 12-24 hour races I've seen have been on shorter loops and/or tracks
Author Response
#REVIEWER 1
Reviewer comment: Overall well done article. Picking 12& 24 races is a novel approach to look at performance characteristics. Comments below
Couple of overall comments:
Reviewer comment: Is it fair to compare men & women given the date ranges for both groups are so different? For women you are looking at a 49 year span and for men you are looking at a 144 year span. Why not just look at data for both over the past 50 years?
Author answer: We appreciate the comment. In fact, between-sex differences exist and it has been well explored in previous studies. These differences are related to a plethora of factors, involving both individual and environmental characteristics. However, please note that the main purpose of the study was not to investigate sex-differences (even though this information was presented mainly in a descriptive view). In addition, regardless of the time range investigated (the last 50 years or even the last 144 years), these differences would be observed, because notwithstanding the female participation had increased in last years, when they started to compete their male peers already had an advantage against them (regarding training, competition experience, support for the practice), and this fact contributes to this sex-differences even when similar number of male and female competitors be observed.
Reviewer comment: You also may not have data on this but could drop out rate/completion rates affect this data. It could be that while finishing times are getting slower, more people are completing the events.
Author answer: Thank you for the comment. We included this point in the discussion section.
Reviewer comment: Abstract: "the sex differences in performance decreased in the last years, when compared to the 32 sex differences in performance during the first years"- could you be more specific with years. Or maybe just say that performance decreased over time?
Author answer: Thank you for the suggestion. We rewrite the sentence.
Reviewer comment: Male runners in 24-hour races were the oldest 37 (46.13±10.83 years), while female runners in the 12-hour races were the youngest (43.46±10.16 years)- Given that range, I'm assuming this wasn't statistically significant. From a reader perspective it's interesting but essentially they are pretty similar age ranges
Author answer: We appreciate the comment. In fact, this is a key point, and we provided more information (methods section) about the age range.
Main Paper:
Intro
Reviewer comment: "This is an invaluable information for both coaches and athletes to prepare training for ultra-endurance events. This information is also crucial for older athletes, due to the decreases in VO2max, lactate threshold velocity, blood volume, and muscle mass". While this is true, I'm not sure how this is "crucial". Most older ultrarunners know they are getting a bit slower
Author answer: Thank you for the comment. We adjusted the sentence, changing the adjective.
Reviewer comment: "between 1876 and 95 2020 for male"- just a comment- I've never reviewed/read a paper that had data from 1876. Fantastic!
Author answer: Thank you for the comment.
Results:
Reviewer comment: Figure 1- Really hard to read given the different dots with same colors. Any reason you can't just do 4 different colored lines?
Author answer: We appreciate the suggestion, and we changed the colors.
Reviewer comment: So the number of people doing these events is increasing. But given you are going all the way back to 1876, the population was much smaller then. In 2021, more people are doing everything. If this were looking at the cost of entering races, you'd have to adjust for inflation. Should you be adjusting for population?
Author answer: We appreciate this information, and we agree with that. However, unfortunately we don’t have this information to perform this adjustment, because it would be necessary not only the population size in each year (or even splinting into periods), but also the percentage of the worldwide subjects involved in regular physical activity practice. It is also interesting to point out (despite the fact this is not the purpose of the present study) that even we have “more people doing everything”, we had never noticed before a such higher number of physically inactive subjects and health related to this behavior in the population. But, taking into account the suggestion proposed, since we do not have the information, we included it as one of the study limitations.
Reviewer comment: For both events, sex differences in performance decreased in the last years, when compared to the sex differences in performance during the first years" Same as abstract comment
Author answer: Sentence adjusted.
Reviewer comment: Figure 2/3- Seems like the X axis should include the years. Also given the men are over a bigger age range, when I look at the graphs, the differences in the x-axis sort of skew directly comparing men's/women's graphs
Author answer: Thanks for the comment. However, we did not include the year in the X axis in the figures because it would be too many information and, visually, it would not look good. Moreover, the purpose of the figures isn’t focusing in each year, but showing the trend across the time. For this reason, we hope the reviewer understands our decision to keep the figures as they were previously presented.
Discussion:
Reviewer comment: "In summary, running events have become a leisure/social activity, with a reduction of the competitive perspective generally observed in the past"- I'm not sure about this. I think in the road marathon situation we've seen this with a big split occurring in the early 90's (the Oprah Winfrey effect in the US). Is there really data that over the past 20-30 years, ultrarunners have become more leisure/social activity? I my experience, I just see more runners getting tired of running on the roads competitively and then they head to the trails. I'd also argue that the depth of the field in elite running has gotten much deeper in the past 10-15 years.
Author answer: We appreciate the comment. Considering non-professional runners, several studies investigating the motivational aspects, for example, have presented that social/health aspects are closely related to running participation.
Reviewer comment: I also would consider thinking about being careful with younger/older categorizations in this paper. (For example_ Athletes ranked in 1st–3 rd positions were the youngest ones (female 12-hour = 260 41.19±8.87 years; female 24-hour = 42.19±8.50 years; male 12-hour = 42.03±9.40 years; male 261 24-hour = 43.55±9.03 years)) Really these aren't young runners in the grand scheme of runners.
Author answer: Thanks for the comment. Indeed, when we consider the age range of the studied sample, the 40’s are not the “youngest” ones. However, please note that the use of these superlatives is done with the purpose to compare a given age group against others.
Reviewer comment: Regarding course/other factors-we are starting to wonder at some of our races if the fastest times may be a thing of the past due to global warming and warmer conditions.
Author answer: This is an interesting point of view.
Reviewer comment: With courses- were the early races on tracks? Do you have a way to separate out results based on the course. While ultrarunning in general is coming up with more and more difficult races, it seems like most 12-24 hour races I've seen have been on shorter loops and/or track.
Author answer: Thank you for the comment. Unfortunately, this information is not available to be
analyzed
Reviewer 2 Report
INTRODUCTION
Please, provide a historic background for ultra-marathon races. When did they started? What were the first distance covered? It has been evolved from a challenge and popular competition to a professionalized event? (I mean, there is a win prize, athletes have to pay a considerable amount of money for inscribing on these races, etc).
Is there a limited time to finish the race? It is a minimum distance that should be covered? It is always the same orography?
It is not clear whether any previous review on this topic has been published, please clarify this point.
I do not see the point on providing hypothesis, since both questions have been previously answered by the literature, as the authors show in this section.
METHODS
I think that there is a need for an inclusion criteria, meaning what is considered an ultra-marathon race. There should be some common characteristics that identify these competitions.
Please, describe who and how collected the data from the website.
RESULTS
I suppose that not all the events shared the same context. Weather factors and orography play and important role on performance and running speed. This should be taken into account when analysing the results.
Similarly, it is plausible to think that running speed decreased due to the increase in popularity of these events, implying that a lot of people are recreational runners. Thus, my advice is to divide the sample of each race in quartiles. I tend to think that when comparing, let’s say the first 50 runners in each competition each year, running speed should have increased since competitive and professional runners are much prepared than years ago, and also material resources and advances in training and physiology have provided better conditions.
DISCUSSION
I think that this section should provide valuable information both for recreational, competitive athletes and sport coaches. How should they approach an ultramarathon race? What is the stimated time that they should employ in the competition taking into account not only age, but they performance level (professional/recreational). This is the key information that manuscript should provide.
Author Response
#REVIEWER 2
INTRODUCTION
Reviewer comment: Please, provide a historic background for ultra-marathon races. When did they started? What were the first distance covered? It has been evolved from a challenge and popular competition to a professionalized event? (I mean, there is a win prize, athletes have to pay a considerable amount of money for inscribing on these races, etc).
Author answer: Information was provided.
Reviewer comment: Is there a limited time to finish the race? It is a minimum distance that should be covered? It is always the same orography?
Author answer: These are time-limited runs where athletes have to run as many kilometers as possible during 12 or 24 hours. We explain this in the method section.
Reviewer comment: It is not clear whether any previous review on this topic has been published, please clarify this point.
Author answer: We included this information in the introduction section.
Reviewer comment: I do not see the point on providing hypothesis, since both questions have been previously answered by the literature, as the authors show in this section.
Author answer: Thanks for the suggestion. We decided to maintain the hypothesis.
METHODS
Reviewer comment: I think that there is a need for an inclusion criteria, meaning what is considered an ultra-marathon race. There should be some common characteristics that identify these competitions.
Author answer: We appreciate the comment. This information was provided in the first sentence of the manuscript: “Ultramarathon races are running events where distances are longer than the traditional marathons, and can be distance (50-km, 100-km, 50-miles, and 100-miles ultra-marathons) - or time-limited (e.g., 6-hour, 12-hour, 24-hour, 48h)”.
Reviewer comment: Please, describe who and how collected the data from the website.
Author answer: We explain in the method section that one of the authors collected all data from the website
RESULTS
Reviewer comment: I suppose that not all the events shared the same context. Weather factors and orography play and important role on performance and running speed. This should be taken into account when analysing the results.
Author answer: In fact, there is a lot of factors that impair runner’s performance. Besides the weather factors and orography, nutritional habits, training background, motivation, physiological factors are related to the performance. However, these data are not available to be included and analyzed in the manuscript. We included these points as limitation of the present study.
Reviewer comment: Similarly, it is plausible to think that running speed decreased due to the increase in popularity of these events, implying that a lot of people are recreational runners. Thus, my advice is to divide the sample of each race in quartiles. I tend to think that when comparing, let’s say the first 50 runners in each competition each year, running speed should have increased since competitive and professional runners are much prepared than years ago, and also material resources and advances in training and physiology have provided better conditions.
Author answer: Thank you for the comment. To consider only the best 50, we would lost statistical power, and we would have to change the study purpose. That is, we are looking for trends in participation and performance across the years, but if we consider only the best 50, we were only able to answer the research question regarding the “elite group”.
DISCUSSION
Reviewer comment: I think that this section should provide valuable information both for recreational, competitive athletes and sport coaches. How should they approach an ultramarathon race? What is the stimated time that they should employ in the competition taking into account not only age, but they performance level (professional/recreational). This is the key information that manuscript should provide.
Author answer: We included this information as practical application.
Round 2
Reviewer 2 Report
The authors have introduced some changes that clarify their manuscript. Nevertheless, they think that there is no need to change the hypothesis and also they do not consider to perform a statistical analysis taking into account percentiles in order to know whether speed have decreased regardless of their athletes’ performance level.
In consequence, the methodological quality of the manuscript has not been sufficiently improved.